# Spermidine-Eugenol Supplement Preserved Inflammation-Challenged Intestinal Cells by Stimulating Autophagy

**DOI:** 10.3390/ijms24044131

**Published:** 2023-02-18

**Authors:** Francesca Truzzi, Anne Whittaker, Eros D’Amen, Maria Chiara Valerii, Veronika Abduazizova, Enzo Spisni, Giovanni Dinelli

**Affiliations:** 1Department of Agricultural and Food Sciences, Alma Mater Studiorum—University of Bologna, 40127 Bologna, Italy; 2Department of Biological, Geological, and Environmental Sciences, Alma Mater Studiorum—University of Bologna, 40127 Bologna, Italy; 3Namangan Engineering Construction Institute, Namangan 160103, Uzbekistan

**Keywords:** autophagy, inflammation, spermidine, eugenol, Caco-2 cells, NCM460 cells, LPS, compound C

## Abstract

Increases in non-communicable and auto-immune diseases, with a shared etiology of defective autophagy and chronic inflammation, have motivated research both on natural products in drug discovery fields and on the interrelationship between autophagy and inflammation. Within this framework, the tolerability and protective effects of a wheat-germ spermidine (SPD) and clove eugenol (EUG) combination supplement (SUPPL) were investigated on inflammation status (after the administration of lipopolysaccharide (LPS)) and on autophagy using human Caco-2 and NCM460 cell lines. In comparison to the LPS treatment alone, the SUPPL + LPS significantly attenuated ROS levels and midkine expression in monocultures, as well as occludin expression and mucus production in reconstituted intestinal equivalents. Over a timeline of 2–4 h, the SUPPL and SUPPL + LPS treatments stimulated autophagy LC3-11 steady state expression and turnover, as well as P62 turnover. After completely blocking autophagy with dorsomorphin, inflammatory midkine was significantly reduced in the SUPPL + LPS treatment in a non-autophagy-dependent manner. After a 24 h timeline, preliminary results showed that mitophagy receptor BNIP3L expression was significantly downregulated in the SUPPL + LPS treatment compared to the LPS alone, whereas conventional autophagy protein expression was significantly higher. The SUPPL shows promise in reducing inflammation and increasing autophagy to improve intestinal health.

## 1. Introduction

Autophagy (meaning “self-eating”) is a physiological cellular process in which damaged cytoplasmic cargo (including long-lived proteins and lipids, misfolded proteins, and organelles) undergo lysosome-mediated self-digestion and recycling to maintain cellular homeostasis [1,2]. Moreover, autophagy improves host defense through several biological functions including the direct elimination of invading pathogens. For this reason, increasing attention has been focused on the role of autophagy in severe Acute Respiratory Syndrome Coronavirus 2 (SARS-CoV-2) replication and pathogenesis [3,4,5,6,7]. Aside from the risk of defective autophagy against pathogens, the shared etiology of defective autophagy and chronic inflammation also constitutes the “hallmarks” of non-communicable diseases including age-associated disorders (cardiovascular, pulmonary, and musculoskeletal diseases, neurodegenerative disorders, cancer, type 2 diabetes, and obesity) as well as auto-immune diseases, collectively representing the leading causes of disability and mortality worldwide [2,8,9,10]. Therefore, research in recent years has been increasingly focused on the complex interrelationship between autophagy and inflammation [2,11].

The best studied aspect involving the direct modulation of autophagy on inflammation is represented by the inhibitory effect of autophagy on inflammasome activation [11,12,13,14,15,16]. Moreover, autophagy modulation of the transcription factor, nuclear factor kappa-light-chain-enhancer of activated B cells (NF-κB), is suggested to afford cellular protection against unrestrained NF-kB activation, responsible for the expression of pro-inflammatory cytokines involved in cellular aging and reduced intestinal epithelial barrier integrity [11,13,17,18]. In this context, the medicinal management of multiple human inflammatory disorders through the modulation of autophagy for therapeutic purposes warrants attention [2,9]. As a result, research on plant-based products has attracted attention in drug discovery fields and testing the efficacy of potential health-promoting products necessitates evaluating autophagy in relation to inflammation.

Interestingly, spermidine (SPD) was cited as one of the most promising candidate autophagy-inducing drugs against “inflammaging” (deterioration of the immune function with age) [10], and future studies aimed at elucidating the precise mechanisms and roles of SPD were proposed to afford unprecedented health benefits [19]. Although SPD is well-reported to be contained in various plant-based foods, wheat germ is the richest source of SPD and is the source of various non-synthetic SPD supplements on the market today, and is also used in experiments on autophagy and inflammaging [20]. Given that wheat-based foods available are predominantly made from refined flour or semolina, wheat germ SPD is not readily accessible. While it is beneficial to consume a varied SPD-rich diet (such as the Mediterranean diet), SPD supplements are recommended in conjunction with a varied diet or in instances where SPD may be deficient (such as in the elderly) or where SPD consumption from local foods is low [20,21]. However, the potential advantage of combining two components with complementary attributes has been highlighted [3]. As part of an incentive of the European Institute for Innovation & Technology (EIT) FOOD (www.eitfood.eu, 1 October 2021), projects using natural products to support either the prevention or treatment of COVID-19, our previous research showed that both wheat germ SPD and clove eugenol (EUG) increased autophagy protein expression and reduced inflammation in vitro [22]. Aside from the anti-viral properties of both components, SPD was selected for reported benefits of increasing autophagy, whereas there is a particular interest in our group in the antioxidant and anti-inflammatory properties of EUG-rich essential oils [23]. Since clove essential oil contains a much higher dose of EUG than whole or ground cloves (which are not widely consumed in European diets), supplements similarly afford the ingestion of EUG in a more concentrated form. Moreover, interest in the use of EUG as a component of medicinal ingredients is currently the object of scientific research [24]. As a result, there was an incentive in developing an economically priced, plant-based, dual component supplement to examine efficacy on both autophagy and inflammation.

As regards the effects of SPD and EUG on the interrelationship between autophagy and inflammation, more research is warranted. For example, the involvement of SPD in the direct autophagy-mediated reduction of inflammation has been shown using either autophagy inhibitors or autophagy-related (ATG) protein knockouts [25,26,27]. Yet, SPD was also shown for the first time in 2022 to directly suppress the NF-κB signaling pathway via the nuclear factor erythroid 2–related factor 2 (Nrf2) that did not involve autophagy [28]. Instead, EUG has been reported to reduce inflammation via NF-κB signaling and inflammasome inactivation [24,29,30], with no reports of the role of autophagy in this process. Only in a single paper was EUG shown to exert a direct attenuating effect on inflammatory ischemia/reperfusion injury through autophagy stimulation [31].

Hence, the first objective was to establish the safety of the combined SPD and EUG supplement (hereinafter referred to as SUPPL) in improving inflammation status and autophagy in vitro utilizing human intestinal epithelium cells as a model. Given that enhanced autophagy is reported to afford protection in the early stages of cancer development but tumor-promoting effects in the more advanced stages [2,32], the non-cancerous intestinal cell line, NCM460, was selected as a representative healthy cell line. An additional objective was to evaluate the effect of the SUPPL on autophagy under both non-inflammatory and inflammatory conditions, and to investigate whether the SUPPL exerts a direct effect on autophagy-mediated inflammation. Since both SPD- and EUG-induced autophagy [25,31] have been reported to occur through the AMP-activated protein kinase/mammalian target of rapamycin (AMPK/mTOR) signaling, the role of autophagy in reducing inflammation was evaluated using dorsomorphin (Compound C), the autophagy inhibitor of AMPK activation.

## 2. Results

### 2.1. Spermidine Composition of the Supplement

In our previous article [22], the increased autophagy protein expression and LC3-II marker staining of Caco-2 cells were evident in response to purified SPD (0.3 mM) extracted from wheat germ and clove essential oil containing EUG (0.2 mM) alone, as well as in a combination treatment (0.3 mM SPD + 0.2 mM EUG). Given the expenses entailed in the extraction of SPD from wheat-germ into pure form together with the microencapsulation of clove essential oil (EUG), producing an economically priced supplement is not feasible. Therefore, a strategy involving the cold extraction of the oil fraction from wheat germ and using the remaining pressed fibrous dry remnant (containing SPD) as a matrix to absorb the EUG was developed as an incentive of one of the European Institute for Innovation & Technology (EIT) FOOD projects. The SPD content in the SUPPL was measured and compared to that contained in the pressed wheat germ matrix (Figure 1). Spermidine was the predominant polyamine detected in both the pressed wheat germ (Figure 1A) and the SUPPL (Figure 1B). The SPD content in the SUPPL was lower (more diluted from the addition of clove essential oil) compared to the pressed wheat germ source (Figure 1C). Putrescine and spermine were present in lower concentrations and collectively constituted approximately half of the total SPD content (Figure 1A,B). Putrescine, SPD and spermine are interconvertible and collectively play a role in stimulating autophagy and protecting intestinal homeostasis [33]. However, given that in wheat germ SPD is the predominant polyamine constituent present, hereinafter the wheat germ component of the supplement will be referred to as SPD. Moreover, given that the powdered matrix of the supplement required dilution prior to administration to the intestinal cell line models, it was necessary to establish the efficacy of a significantly lower SPD content on autophagy compared to the SPD concentration in pure form used previously [22].

### 2.2. Effects Spermidine and Eugenol Treatments on Autophagy LC3-II Marker Detection and Tolerability in Caco-2 and NCM460 Cell Lines

The powdered SUPPL wheat germ matrix, diluted to a 1.2 µM SPD content with a corresponding absorbed EUG content of 92 µM, was prepared. The first objective was to investigate LC3-II marker staining in response to the ready-prepared supplement (principal components: 1.2 µM SPD and 92 µM EUG) in Caco-2 lines compared to the higher pure form SPD contents reported previously [22]. The steady state lipidified protein 1 light chain 3 (LC3-II, or LC3B), produced from the covalent linkage of cytosolic LC3-I to the phosphatidylethanolamine (PE) lipid on the surface of the phagophore, was firstly employed as an indirect marker of increased autophagy. A 4 h exposure time of different treatments was administered to Caco-2 cells as SPD alone (pressed wheat germ, 1.2 µM SPD), EUG alone (200 µM), and then the combined treatments. The first combined treatment was performed by adding SPD (1.2 µM) + EUG (200 µM) separately (hereinafter: SPD + EUG) as reported previously [22]. This was compared to the second combined treatment, namely the SUPPL.

In all treatments, LC3-II red chromogen marker staining was shown to be increasingly evident in the Caco-2 cells compared to the untreated control (Figure 2A). From the red chromogen images (Figure 2A), the percentage of LC3-II stained cells in response to each treatment was calculated (Figure 2B). Compared to the untreated control, with approximately 40% LC3-II stained cells, there was a significant and equivalent increase to approximately 80% LC3-II stained cells after a 4 h exposure to all treatments, respectively (Figure 2B). The starvation treatment (a well-documented positive control to induce autophagy) showed a similar increase. As a percentage of the untreated control, 1.2 µM SPD was inferred to have stimulated autophagy in Caco-2 cells to an equivalent extent demonstrated previously for 300 µM SPD [22].

Since enhanced autophagy is reported to demonstrate tumor-promoting effects in the more advanced stages of cancer [32], and that cancer cell lines demonstrate differing responses to autophagy modulators compared to non-cancer cell lines [34,35], a normal human colon epithelial cell line NCM460 was also selected to investigate the efficacy of the SUPPL in comparison to the standard colorectal adenocarcinoma Caco-2 model [22]. The percentage of LC3-II stained cells was increased to a comparable extent in NCM460 cells compared to the Caco-2 cells after exposure to all treatments (Figure 2C). The starvation control was, similarly, shown to increase LC3-II staining in NCM460 cells.

Given the comparable efficacy in stimulating LC3-II of the premade SUPPL in comparison to the combination treatment of active ingredients (SPD + EUG administered separately), the safety and tolerability of the SUPPL were then investigated. From the 3-(4,5-dimetiltiazol-2-il)-2,5-difeniltetrazolio (MTT) assay, there were no toxic effects from the SPD, EUG, SPD + EUG, and the SUPPL treatments, respectively, on the NCM460 cells, all showing a comparable percentage to the untreated control after 24 h (Figure 3A). Of relevance, the administration of the SUPPL was not shown to induce a significantly different effect on cell viability compared to the control in either the standard Caco-2 cell line or the NCM460 human colon epithelial cell line (Figure 3B).

### 2.3. Effects of Spermidine and Eugenol Treatments on Autophagy LC3-II Marker Turnover in NCM460 Cell Lines in the Presence and Absence of Liposaccharide

As the association between autophagy and inflammation in response to plant-based modulators (including SPD) is a developing field of study [2,11,24,25,26,27,36,37], the next objective was to examine the effect of the SUPPL in the absence and presence of inflammation. To induce inflammation, a concentration of 1 ng/mL LPS was selected. This is within the range of physiologically and clinically relevant concentrations (0.3–10 ng/mL) previously reported to promote oxidative stress and intestinal barrier permeability, but not cell death [38,39]. This LPS content was previously used by our group to induce inflammation [22].

A direct approach in estimating autophagic flux was used with the late autophagy blocker, hydroxychloroquine (HCQ) [40]. To this end, LC3-II flux was firstly visualized using fluorescent green (FG) in the control, and in response to the administration of LPS, the SUPPL, and LPS + SUPPL, respectively, in both the presence and absence of 20 µM HCQ for a short timeline of 2 h (Figure 4A). In addition, LC3-II, as well as sequestosome-1 (also known as the ubiquitin-binding protein P62) turnover in both the presence and absence of HCQ was visualized in all treatments using red chromogen staining (Figure 4A), and then statistically quantified (Figure 4B,C).

Both the untreated control and LPS treatment alone after 2 h produced basal levels of FG LC3-II labeling (Figure 4A). Compared to the treatments without HCQ, the addition of HCQ to the control and LPS treatment resulted in an increase in LC3-II puncta (Figure 4A), resulting from a block in the degradation of cargo in LC3-II labeled autophagosome-lysosome fusion [40]. The administration of the SUPPL alone and the SUPP + LPS alone resulted in a very evident accumulation of LC3 puncta over a 2 h period (Figure 4A). Then, in combination with HCQ, the number of LC3-II puncta increased and became fused (Figure 4A). This accumulation far exceeded any accumulation of puncta resulting from the HCQ block at basal levels (control and LPS treatment).

Red chromogen imaging was then also performed with the objective of quantifying LC3-II expression. The levels of LC3-II in the CTRL + HCQ and LPS + HQC were equivalent and significantly lower in comparison to the SUPPL + HCQ and SUPPL + LPS + HCQ treatments, respectively (Figure 4B). LC3-II levels in the absence of HCQ were significantly lower than those in the presence of HCQ for all treatments (Figure 4B). LC3-II levels in the SUPPL + LPS and SUPPL + LPS + HCQ were significantly higher in the remaining treatments (Figure 4B). P62, which binds to LC3, was also used to monitor autophagic flux (Figure 4A,C). As with LC3-II, P62 levels in the absence of HCQ were significantly lower than those in the presence of HCQ for all treatments (Figure 4C). P62 expression in the LPS and LPS + HCQ treatments was more evident compared to that of LC3-II in the same treatments (Figure 4B,C). In contrast to LC3-II expression, which was higher in the SUPP + LPS and SUPP + LPS + HQC treatments compared to the SUPP and SUPP + HCQ treatments, respectively, P62 was equivalent between respective treatments (Figure 4B,C).

### 2.4. Effect of the Spermidine and Eugenol Combination and Supplement Pretreatments on the Expression of Autophagy Microarray Expression in NCM460 Cells Treated with Lipopolysaccharide

As multiple, parallel methods to investigate autophagy are recommended [40], and given that autophagy is a multistep pathway, the proteomic profiles (Figure 5A) of 20 human autophagy proteins spanning all stages of autophagy were examined (Figure 5B) and analyzed statistically (Figure 5C) to indirectly evaluate the effect of LPS, LPS + SPD + EUG, and LPS + SUPPL after a longer period of 24 h.

Compared to the untreated control, there was a significant increase in BCL2/adenovirus E1B 19 kDa protein-interacting protein 3-like (BNIP3L, commonly referred to as NIX) in the LPS-treated cells (Figure 5C). BNIP3L, an inducer of mitophagy (a selective autophagy pathway that targets mitochondria to maintain mitochondrial quality control) directly binds to either LC3-II or γ-aminobutyric acid A receptor–associated protein (GABARAP) homologues to recruit dysfunctional mitochondria in the autophagasome [41,42,43]. In the LPS-treated cells, increased BNIP3L was associated with increased GABARAP expression but not LC3-II (Figure 5C). Compared to the untreated control, in the LPS-treated cells there was also an increase in P62 and Beclin-1 (Figure 5C). An associated increase in ATG10, important in autophagosome formation, was also evident (Figure 5C).

Pre-incubation with SPD + EUG and the SUPPL, respectively, for 1 h prior to LPS addition was then examined on autophagy protein expression after a 24 h exposure (Figure 5). The inclusion of both SPD + EUG and the SUPPL significantly decreased mitophagy-inducer BNIP3L expression to a comparable extent (Figure 5C). Moreover, both SPD + EUG and SUPPL pretreatments significantly increased various conventional autophagy-related (ATG) proteins compared to the untreated control at all stages of the autophagy pathway including: ATG3, ATG4A, ATG4B, ATG5, ATG10, ATG12, ATG13, Beclin-1, LC3-II, GABARAP, and lysosomal associated membrane protein 1 (LAMP1), respectively (Figure 5C). Pre-incubation with SPD + EUG and the SUPPL showed equivalent levels of ATG4A, ATG4B, ATG10, ATG13, Beclin-1, and GABARAP, whereas pre-incubation with SPD + EUG resulted in a significantly higher expression of ATG3, ATG5, ATG12, and LAMP1 compared to the SUPPL (Figure 5C). Interestingly, ATG7, required for LC3-I lipidation along with ATG3 and ATG5, was significantly reduced in the treatments compared to the untreated control (Figure 5C). The increased presence of LC3-II after 24 h in both the SPD + EUG and SPD-EUG treatments may suggest that ATG7 was not limiting but became depleted.

### 2.5. Effect of Spermidine and Eugenol Pretreatments on Inflammatory Parameters in NCM460 Cells Treated with Lipopolysaccharide

The efficacy of the SPD and EUG plus SUPPL pretreatments in reducing inflammatory parameters was investigated. A pretreatment with SPD, EUG, SPD + EUG, and the SUPPL, respectively, for 1 h prior to the addition of LPS for 24 h resulted in improved NCM460 cell proliferation using the MTT assay, compared to the LPS treatment alone (Figure 6A). Midkine (MDK), an endogenous inflammatory marker induced by the NF-κB pathway, also associated with inflammatory diseases of intestinal cells [44], was then measured to assess the effect resulting from the pretreatment with SPD and EUG under LPS exposure. The human MDK ELISA kit, containing a precoated MDK antibody, was used to bind MDK in the samples, which was then quantified. There was no significant difference in MDK levels between the untreated control and the cells exposed to all the treatments for 24 h in the absence of LPS (Figure 6B). The LPS treatment alone induced a significant increase in MDK levels after 24 h, exceeding the expression levels in the untreated control. Preincubation with SPD + EUG and the SUPPL for 1 h, prior to the addition of LPS, was shown to reduce MDK expression to the level of the untreated control (Figure 6B).

Lipopolysaccharide-induced oxidative stress was also assessed by measuring reactive oxygen species (ROS) production, since LPS treatment is widely reported to induce increased production of ROS as well as mitochondrial dysfunction. The Human Reactive Oxygen Species Modulator 1 (ROMO1) ELISA Kit was used to quantify ROS based on the extent of binding to the ROMO1 antibody. Reactive oxygen species was marginally but significantly higher after exposure to LPS for 24 h compared to the untreated control (Figure 6C). Pre-incubation with the SUPPL for 1 h prior to LPS addition significantly reduced ROS accumulation to the levels of the untreated control. Instead, pre-incubation with the SUPPL prior to LPS addition significantly reduced the ROS levels only when compared to the LPS treatment alone (Figure 6C).

### 2.6. Effect of Spermidine plus Eugenol and Supplement Pretreatments in Reconstituted Caco-2 Intestinal Equivalents Exposed to Liposaccharide

Since inflammation in intestinal cells is widely reported to result in the expression of pro-inflammatory cytokines that impact negatively on TJ protein expression and mucus production, it was necessary to use more physiologically and structurally relevant in vitro models to investigate these parameters. For this reason, a three-dimensional (3D) co-culture system (Caco-2/U937/L929 cells) was constructed with which to investigate the effect of both SPD + EUG and the SUPPL pretreatments on cellular integrity, LC3-II expression as well as occludin and mucus expression under LPS-mediated inflammation. As Caco-2 cell lines are widely used in the standard construction of 3D models, the latter were utilized in the present study.

Hematoxylin and eosin (H&E) staining showed the preservation of the Caco-2 columnar cells forming a tight and regular monolayer in both the control as well as the cells pre-treated for 24 h with both SPD + EUG and the SUPPL, respectively, prior to the addition of LPS for 24 h (Figure 7A). The disruption of the monolayer was evidenced after a 24 h exposure to LPS alone (Figure 7A). LC3-II expression was significantly and equivalently higher in both SPD + EUG- and SUPPL-treated cells prior to LPS addition compared to the untreated control (Figure 7B). The untreated Caco-2 control cells displayed comparable LC3-II expression to the LPS-treated cells (Figure 8B), corroborating our previous research [22]. Occludin, forming an integral component of the TJ proteins, was shown to be significantly decreased in LPS-treated cells compared to the untreated control (Figure 7C). The 24 h pre-treatment with both SPD + EUG and the SUPPL prior to incubation with LPS for 24 h significantly restored occludin protein expression compared to the LPS-treated cells alone (Figure 7C). Although Caco-2 cells cultured alone are comparatively low in mucus expression compared to intestinal epithelial models comprised of Caco-2 and HT29 co-cultures [45], Caco-2 cells were nonetheless investigated for mucus production in the present study. The LPS treatment alone significantly stimulated mucus production compared to the untreated control (Figure 7D). The SPD + EUG and the SUPPL pretreatments, followed by the 24 h exposure to LPS, significantly reduced mucus expression compared to the LPS treatment alone, although not to the same level as the control (Figure 7D).

### 2.7. Effect of the Supplement Pretreatments on the Expression of LC3-II Marker Detection and Midkine Expression in NCM460 Cells Treated with Lipopolysaccharide and Compound C

After examining the effect of the SUPPL on autophagy (under both the absence and presence of LPS) as well as on inflammation parameters, the next objective was to investigate the direct effect of conventional autophagy on reducing inflammation by inhibiting AMPK activation using the early autophagy blocker Compound C (CC). Compound C at the concentration of 20 µM was previously shown to block autophagy in healthy NCM460 cell lines after 2 h [35]. The choice of Compound C was based on previous results suggesting that both SPD and EUG stimulate autophagy at AMPK/mTOR [25,31].

The presence/absence of autophagy was assessed from red chromogen images (Figure 8A) from which the percentage of LC3-II stained cells was calculated (Figure 8B) and compared to the inflammation status using MDK as a marker (Figure 8C). All treatments that included 5 µM CC (incubated for 30 min prior to the addition of the supplement) exhibited a significantly lower LC3-II expression compared to the basal levels in the untreated control (Figure 8A,B). The integrity of the cells was not compromised by the addition of CC (Figure 8A). Addition of the SUPPL alone, or in a pretreatment for 1 h prior to a subsequent 4 h exposure to LPS, showed a significantly increased expression of LC3-II stained cells (Figure 8A,B) confirming previous results (Figure 2A,C, Figure 4 and Figure 5A,C).

The block in autophagy LC3-II expression by CC alone did not induce an increase in MDK levels (Figure 8C), which were comparable to the control. The addition of LPS alone for 4 h resulted in significantly higher MDK levels compared to the basal untreated control levels. Although MDK levels after a 4 h exposure to LPS were significantly lower than those reported after a 24 h exposure (compare Figure 8C with Figure 6B), a 4 h period was sufficient to induce significant inflammation. The addition of the SUPPL for 1 h to cells that had been incubated with CC for 30 min, followed by a 4 h exposure to LPS, showed a decrease in MDK, suggesting that MDK reduction was not directly regulated by autophagy within this timeframe.

To investigate whether the autophagy-dependent reduction of inflammation became important over a longer-term period, the preceding experiment was repeated using an LPS exposure of 24 h. However, 5 µM CC alone was shown to significantly reduce cell viability after a 24 h period, showing that the complete blockage of autophagy was detrimental (Appendix A).

## 3. Discussion

The shared etiology of defective autophagy and chronic inflammation constitute the “hallmarks” of many non-communicable and auto-immune diseases, for which research on natural products has attracted attention in drug discovery fields. This interest is paralleled with investigating the effects of products on the interrelationship between autophagy and inflammation [2,11,25,26,27,28,36,37]. Within this framework, the tolerability and protective effects of an economically priced SUPPL, composed of pressed wheat germ matrix containing SPD with adsorbed clove EUG, on inflammation status and autophagy were investigated in intestinal Caco-2 and NCM460 cell lines.

The safety and tolerability of the SUPPL on both Caco-2 and healthy NCM460 intestinal epithelial cells as preclinical material sources were shown, since no toxic effects on cell proliferation (MTT assay) were evident. Moreover, the supplement was also prepared taking into consideration the non-observed adverse effect level (NOAEL) for both components, as well as factoring in acceptable human safety factors. Both the SPD and EUG components of the SUPPL were within the safety limits set for human consumption [20,23,46], and by inference also for human cell lines in the concentrations used. Given the current interest in the use of EUG as a component of medicinal ingredients [24], the need for future studies involving the selection of specified doses for various functional applications was highlighted [47,48]. Using the selected concentrations in the SUPPL, the anti-inflammatory efficacy of EUG in combination with SPD is firstly addressed.

The SUPPL was shown to ameliorate inflammation induced by LPS. The administration of LPS alone to the intestinal cells induced significantly higher ROS production and NF-kB-inducible MDK expression, thereby corroborating extensively documented reports of LPS stimulating mitochondrial ROS and pro-inflammatory cytokine production. The exposure of the 3D reconstituted models to LPS alone was shown to decrease TJ occludin content and increase mucus production compared to the untreated control. These results similarly supported previous reports showing both a decrease in TJ proteins [49,50], and an increase in mucus production [51] induced by NF-kB activation, even at physiologically relevant concentrations of LPS [38,39]. The pretreatment of the cells with the SUPPL significantly ameliorated ROS production and MDK expression. Both the SPD and EUG components of the SUPPL have been previously reported to reduce oxidative stress and NF-kB-inducible pathways [23,25,26,27,29,30,47,48]. The SPD + EUG and SUPPL treatments attenuated the effect of LPS by restoring occludin protein content, also corroborating previous findings on SPD [30,52] and EUG [53,54] restoring TJ proteins, thereby improving barrier integrity. This is particularly important for intestinal cells, where barrier permeability is a hallmark of various gastrointestinal diseases [9,13]. Furthermore, the SPD + EUG combination and the SUPPL treatments also attenuated the effect of LPS by decreasing excessive mucus secretion. Although Caco-2 cells, used in the 3D model, are reputed to be low-mucus-producing cells [45], Caco-2 cells are reported to express mucins MUC5A/C [55] under inflammatory conditions.

Given that inflammation has been shown to be inextricably linked to autophagy, autophagy in response to the SUPPL was addressed. Despite the increasing knowledge about the role of autophagy in modulating inflammation, this field is still only at its beginnings and more research is required on different cell types over different timelines and for different potential therapeutic compounds, respectively [2]. Interestingly, in recent reviews on the anti-inflammatory/antioxidant activities of EUG, there is no mention of the role of autophagy [24,29,30,47,48]. As regards the SPD component, published results are still scarce on this topic. Within a 24 h timeline, autophagy has either been directly implicated in modulating inflammation [26,27], or not [28]. In the present study, over a short timeline of 2–4 h, the SUPPL was shown to significantly increase steady state LC3-II levels compared to the untreated control. LC3-II expression was similarly increased under starvation, commonly used as a positive control of increased autophagy. The pretreatment of the SUPPL under LPS-induced inflammatory conditions (evidenced by elevated MDK) increased LC3-II expression to higher levels than with the SUPPL alone. LC3-II turnover, in the untreated control and LPS treatments alone following a block in autophagy with HQC, evidenced minimal levels of autophagic flux over 2 h. In contrast, LC3-II puncta in the presence of either the SUPPL alone or SUPPL + LPS, both without and with HCQ, significantly exceeded basal levels evident in the untreated control and LPS treatment. From the quantification of LC3-II and P62 expression in the presence of HCQ, a stimulation of protective autophagy by the SPD and EUG components of the SUPPL under both non-inflammatory and inflammatory conditions was evidenced from these parallel autophagy measurements. The results corroborated previous studies verifying an increase in autophagic flux by SPD in immune and cardiac cell lines using chloroquine [25,26,27].

Still within the short timeline, given that the SUPPL induced autophagy and reduced inflammation, the question that arose was whether (and to what extent) the SUPPL acted to reduce inflammation by the direct modulation of autophagy. We attempted to address this question by examining the effect of the SUPPL in reducing LPS-induced inflammation in NCM460 cells in which AMPK-mediated autophagy was effectively blocked using CC (indicated by the absence of LC3-II expression) over a period of 4 h. In the absence of autophagy (CC-treated NCM460 cells), the SUPPL was shown to reduce MDK expression, suggesting that, over a short timeline, non-autophagy-dependent antioxidant pathways may have been activated to reduce inflammation. Interestingly, in a recently published paper, SPD was shown to inhibit the LPS-induced pro-inflammatory activation of murine-derived macrophages by acting on Nrf2 signaling but not autophagy when blocked with the late autophagy blocker bafilomycin for 24 h [28]. These results contrasted with previous results also using murine-derived immune cells where SPD was shown to attenuate inflammation in an autophagy-dependent manner [26,27]. Interestingly, given that an interaction between EUG and Nrf2-signaling has also been demonstrated [56], the effect of the SUPPL on Nrf2 signaling warrants further investigation. The autophagy blockage of healthy NCM460 cells for a period of 24 h by CC alone was shown to compromise cell viability and the role of autophagy on modulating inflammation could not be determined.

Over a 24 h period, LPS + SPD + EUG and LPS + SUPPL treatments increased the steady state expression of conventional autophagy proteins, which were largely comparable to those shown over a shorter 4 h timeline with SPD + EUG alone [22]. The increases in autophagy proteins were involved in all stages of autophagy, which included: the initiation of the double membrane vesicles (ATG13) [57], nucleation to form phagophores (Beclin-1) [5], the elongation of phagophores to form autophagosomes (ATG3, ATG4A, ATG4B, ATG5, ATG10, ATG12, LCB3-II, GABARAP) [4,5,6], and finally the fusion of autophagosomes to the lysosomes (ATG13; Beclin-1, LAMP1) [5,57,58,59]. Whether the increases in conventional autophagy proteins reflected an increase in protective autophagy remains to be verified using late autophagy blockers. Of interest, SPD + EUG and the SUPPL, in the presence of LPS, attenuated only the expression of BNIP3L, which was significantly expressed in the LPS treatment alone. BNIP3L is currently recognized as a key regulator of mitophagy [41,42,43,60,61], and whether the downregulation of this mitophagy receptor was attributable to the SUPPL-induced activation of antioxidant pathways resulting in the reduction of ROS (and inflammatory MDK) under LPS challenge remains to be determined.

The SUPPL was shown to be equally efficient in reducing inflammatory parameters and increasing autophagy protein expression as when the two components were added separately (SPD + EUG), showing that in a SUPPL form the principal ingredients retained their properties. Although a healthy, varied diet is always recommended, the SUPPL was shown to afford protection to intestinal cells and may be used in conjunction with a healthy diet, as well as for individuals with inflammatory issues. Although the present study is preliminary, to the best of our knowledge there are no previous studies using a preprepared dual component SUPPL specifically addressing the relationship between autophagy and inflammation.

## 4. Materials and Methods

### 4.1. Materials

In conjunction with the University of Bologna, the Targeting Gut Disease (TGD) company (https://www.tgd.care/company, 1 October 2022) developed the SPD-EUG supplement. Pressed “defatted” dried, milled wheat germ (a by-product of industrial wheat germ oil extraction) was both the source of SPD as well as the “support” to adsorb and stabilize EUG. Within a 550 mg supplement capsule, SPD was contained in 333 mg pressed wheat germ (approximately 0.17 mg SPD, and approximately 0.055 mg of putrescine and spermine) onto which 67 mg of essential oil from clove (containing 60.3 mg EUG) was absorbed. Although, the supplement was administered to human cell lines in the present study, the safety of the supplement was nonetheless determined. The non-observed adverse effect level (NOAEL) for EUG is 300 mg/kg body weight per day, corresponding to 21 g/day for an individual weighing 70 kg [23,46]. After factoring in a safety factor of 70 for human consumption, the EUG content in the supplement was considered within the limit of safety for human consumption and therefore by extension also for human cell lines in the concentrations used (see below). Similarly for SPD, polyamine-rich extract (wheat germ) for human consumption is calculated at 41 mg/kg bodyweight per day or 2.8 g extract (containing circa 1.5 mg spermidine) for the average person weighing 70 kg [20]. Hence, the SPD levels were also within the safety limit. The pressed wheat germ was used as the source material for the administration of SPD alone or in combination with exogenously added EUG (SPD + EUG). Essential oil of EUG was provided from TGD, and similarly diluted in ethanol with purity of 99.5% for the administration of EUG alone or in combination with SPD.

Reagents for cell cultures, such as Dulbecco’s Modified Eagle Medium (DMEM), Roswell Park Memorial Institute Medium (RPMI), Fetal Bovine Serum (FBS), L-Glutamine, Penicillin-Streptomycin, and rat tail collagen type I were purchased from GIBCO (Waltham, MA, USA). Hydroxychloroquine sulfate was purchased from Merk-Sigma Aldrich (H095). The MTT was from Life Technologies (Carlsbad, CA, USA) and Compound C (Dorsomorphin) from APExBIO Technology (Houston, TX, USA). LC3-II and occludin antibodies were from Novus Biologicals (Milan, Italy), whereas the SQSTM1/P62 was from ABclonal Technology (Düsseldorf, Germany). The UltraTek Alk-Phos Anti-Polyvalent (permanent red) Stain Kit and Alcian Blue/PAS kit were purchased from ScyTek Laboratories, Inc. (Logan, UT, USA). The Alexa Fluor 594-conjugated goat IgG was from Thermo Fisher Scientific (Waltham, MA, USA) and H&E from Bio-Optica^®^ (Milan, Italy). The Human Reactive Oxygen Species (ROS) ELISA Kit and the Human Midkine ELISA kit were from ABclonal Technology (Düsseldorf, Germany) and Cohesion Biosciences (Suzhou, China), respectively. The Human Autophagy Array C1 [AAH-ATG-1] (RayBio^®^ C-Series) was from RayBiotech (Norcross, GA, USA). All other chemicals and solvents were of analytical grade.

### 4.2. Polyamine Extraction and Measurement

The powdered wheat germ matrix alone and that used for the supplement were, respectively, extracted for polyamine content using an acid water solution containing 0.4% (*v*/*v*) trifluoroacetic acid (TFA). The material containing the soluble polyamine content was sonicated for 15 min and then filtered at 0.22 µm. The polyamines (SPD, putrescine, and spermine) were separated and analyzed by HPLC-ESI-MS (Waters e2695 equipped with Acquity QDa detector) using a Luna Omega polar C18 column (Phenomenex, Torrance, CA, USA), 5 µm, 250 × 4.6 mm set at a temperature of 30 °C. The mobile phase was composed of 15% methanol, 5% acetonitrile, and 80% TFA solution (0.4% in water), respectively. Detection parameters were as follows: positive ionization, 12 V cone voltage, and probe temperature of 600 °C.

### 4.3. Cell Model Systems and Growth Maintenance Conditions

NCM460, a normal human colon mucosal epithelial cell line (RRID: CVCL_0460), purchased from BeNa Culture Collection (Shanghai, China), was cultured in DMEM containing 10% FBS, 1% penicillin-streptomycin, and 1% L-glutamine. The Caco-2 human epithelial cell line (ATCC HTB-37), obtained from colorectal adenocarcinoma, was cultured with DMEM, supplemented with 10% FBS and 1% penicillin-streptomycin. L929 mouse fibroblasts (ATCC-CCL1) and U937, a pro-monocytic human myeloid leukemia cell line, (ATCC CRL-1593.2) were only included in the 3D co-cultured models. L929 cells were cultured with DMEM, composed of 10% FBS, 1 mM L-glutamine, and 1% penicillin-streptomycin, whereas U937 cells were cultured in RPMI-1640 medium, supplemented with 10% FBS and 1% penicillin-streptomycin. Stock cultures of all cell lines were maintained at 37 °C in a humidified atmosphere containing 5% CO_2_ in tissue culture flasks (75 cm^2^; BD Biosciences), and the culture medium changed every two days. Prior to experimentation, Caco-2 and NCM460 cells were trypsinized and cell density evaluated microscopically using a Bürker counting chamber.

### 4.4. Experimental Monoculture Conditions to Investigate Spermidine and Eugenol Treatments in the Absence or Presence of the Inflammation Elicitor Lipopolysaccharide

NCM460 (1 × 10^5^ cells/well) and Caco-2 cells (1 × 10^5^ cells/well) in complete medium were, respectively, plated onto 96-well plates and incubated for 24 h. Thereafter, concentrations of SPD and EUG both alone, in combination, and in SUPPL form, respectively, were diluted in DMEM containing 10% Bovine Serum Albumin (BSA). The standard concentrations administered to the cells were as follows: 1.2 µM SPD alone, 0.2 mM EUG alone, the combination treatment of 1.2 µM mM SPD + 0.2 mM EUG, and the SUPPL (1.2 µM SPD and 92 µM EUG). The untreated control for each cell line contained only the culture medium and 10% BSA, whereas the starvation control contained complete medium without 10% BSA. Following a 24 h exposure to the treatments, the medium was carefully aspirated, and cell proliferation (MTT assay) measured.

For the autophagy LC3-II marker experiments, Caco-2 and NCM460 lines (1 × 10^5^ cells/well) in complete medium (DMEM,) were, similarly, respectively, plated into 4-well chamber slide plates and incubated for 24 h. Thereafter, each cell line type was exposed to the above-mentioned SPD and EUG treatments, respectively. Controls (untreated control and starvation control) were also included. After a 4 or 24 h exposure, cells were fixed for LC3-II marker detection.

For the autophagy flux measurements using LC3-II and P62 as markers in the presence/absence of HQ and/or LPS, respectively, NCM460 (1 × 10^5^ cells/well) cells were seeded in complete medium onto 8-well chamber slide plates. After 24 h, the medium was removed and replaced with fresh medium containing either containing 20 µM HQC or no HQC. Either DMEM (untreated control) or the SUPPL or LPS (1 µg/mL) alone and in combination were added to the cells for a period of 2 h.

For the autophagy LC3-II marker and inflammation MDK marker experiments in the presence/absence of CC and/or LPS, respectively, NCM460 (1 × 10^5^ cells/well) cells were seeded in complete medium onto 8-well plates. After 24 h, the medium was removed and replaced with fresh medium containing either 5 µg CC, or without CC. After 30 min, either the SUPPL or DMEM (to the untreated control or LPS treatment alone) were added to the cells for a period of 1 h. Then, 1 ng/mL LPS was added to the LPS treatment alone as well as to the experimental treatments containing the SUPPL, with or without CC, for a further 4 h. Thereafter, the supernatants were collected for MDK measurements, and the cells were fixed for LC3-II marker detection.

For the cell proliferation, ROS and MDK measurements in the presence of LPS, NCM460 (1 × 10^5^ cells/well) in complete medium (DMEM) were plated into 96-well chamber slide plates and incubated for 24 h. NCM460 cells were similarly pre-treated with SPD alone, EUG alone, the combination treatment, and the SUPPL, respectively, for 1 h. Thereafter, 1 ng/mL LPS was added, and the cells incubated for a further 24 h before carrying out the respective analyses. The untreated control, which contained no LPS or treatment substrates, was made up to the same volume using DMEM. Cells treated with LPS alone (1 ng/mL in DMEM) without the pretreatments were also included.

### 4.5. Cell Proliferation (MTT) Assay

Proliferative Caco-2 and NCM460 cells in monoculture were detected using the MTT assay, according to the ISO 10993-5 International Standard procedure (ISO 10993-5, 2009). The method is based on the reduction of MTT by mitochondrial dehydrogenase of intact cells to produce purple formazan, determined by measuring the absorbance at 540 nm using a multiwell scanning spectrophotometer (Labsystems Multiskan MS Plate Reader, ThermoFisher Scientific, Waltham, MA, USA), as described by Truzzi et al. [22]. Results were expressed as the percentage of viable cells with respect to untreated controls (70% ethanol). The percentage of cell proliferation was calculated using the following formula: absorbance value of treated sample/absorbance value of control × 100 = % of cell viability.

### 4.6. Immunocytochemistry for Autophagy LC3-II Marker Detection

Caco-2 and NCM460 cell lines, plated and treated in the four well Permanox chamber slides, were fixed in 70% ethanol for 10 min. Cells were permeabilized with 0.1% TritonX100 and stained for LC3-II (Novus, St. Louis, MO, USA) antibody and then labeled with the UltraTek Alk-Phos Anti-Polyvalent (permanent red) staining kit according to the manufacturer’s instructions. Then, slides were stained with haematoxylin, which stains the nuclear material deep purple. The slides were examined under the microscope (MEIJI Techno Co., Ltd., San Jose, CA, USA) at a magnification of ×40 to identify LC3-II positive cells. The nuclei were quantified (representative of Figure 2), and more than 400 nuclei were counted for each sample. Results were expressed as the percentage of LC3-II positive cells to the number of nuclei counted.

### 4.7. Immunofluorescence and Immunocytochemistry for Autophagy LC3-II and P62 Expression and Turnover with Hydroxychloroquine

NCM460 cells were fixed with 4% paraformaldehyde for 15 s and fixed in ice-cold 100% methanol for 10 min at −20 °C. For the immunofluorescence staining, cells were permeabilized with 0.1% TritonX100 and stained for LC3-II (Novus) antibody and then labeled with Alexa Fluor 594-conjugated goat IgG for FG imaging, according to the instructions provided by the manufacturer. Nuclear staining was performed for DAPI (4′,6-diamidino-2-phenylindole), a blue-fluorescent DNA stain used to stain the nuclear material in FG imaging. For the immunocytochemistry staining, cells were stained for LC3-II (Novus) antibody and SQSTM1/P62 (ABclonal) antibody, respectively, and labeled with the UltraTek Alk-Phos Anti-Polyvalent (permanent red) staining kit according to the manufacturer’s instructions. The nuclear material was stained purple with haematoxylin. LC3-II and P62 expression on slides stained for immunocytochemistry were quantified as follows. Pictures of cells were analyzed by using ImageJ2 software (Wayne Rasband, version 2.9.0/1.53t). To perform the analysis of the pixels, digital images were processed to 300 pixels/inch and converted to 8 bits. Then, the binary images were subjected to “color deconvolution” plugin to analyze permanent red staining. The selected picture was saved as a tiff and subjected to a “clean-up” procedure to eliminate artefacts with Adobe PhotoshopCC (version 20.0.4). Thereafter, the area of interest was measured with the application “Analyze particle” of ImageJ2. Each experiment was performed in triplicate and data reported as the number of pixels.

### 4.8. ROS and Midkine Detection

Reactive oxygen species was measured using the ROMO1 ELISA Kit according to the instructions provided in the kit. In brief, samples were added to wells of a microplate precoated with ROMO1 antibody. Reactive oxygen species present in the sample and standards were then bound by the immobilized antibody, after which an enzyme-linked polyclonal antibody specific for ROMO1 was added. The complex was visualized from the addition of a color-based substrate. The amount of ROS present was proportional to the intensity of the color measured at 450 nm. Midkine in the supernatants was measured using the human MDK ELISA kit. The principle employed an antibody specific for Human MDK coated on a 96-well plate. Midkine present in the samples and standards was bound to the wells by the immobilized antibody. Biotinylated anti-Human MDK antibody was then added and subsequently bound to HRP-conjugated streptavidin. Finally, a substrate was added, and the color measured at 450 nm. Color intensity was proportional to the amount of bound MDK.

### 4.9. Detection of Autophagy Protein Activation

The Human Autophagy Array C1 [AAH-ATG-1] (RayBio^®^ C-Series) for the semi-quantitative detection of 20 human proteins in cell and tissue lysates was used to measure autophagy protein activation. NCM460 cells (1 × 10^5^ cells/well) in complete medium were plated onto 4-well chamber slide plates. After 2 h, cells were exposed for 1 h to either the combination treatment of 1.2 µM mM SPD + 0.2 mM EUG, and the SUPPL (1.2 µM SPD and 92 µM EUG), respectively, or to DMEM (untreated control and LPS treatment alone). After 1 h, 1 ng/mL LPS was added to the LPS treatment alone, the SPD + EUG combination treatment, and the SUPPL, respectively, and incubated for a further 24 h. Proteins were extracted from each of the treatments, and 200 µg was added to Antibody Array membranes and detected according to the manufacturer’s instructions.

### 4.10. Autophagy LC3-II Marker, Occludin, and Mucus Detection in a Reconstituted Intestinal Cell Model with Caco-2, U937, and L929 Cells

The 3D reconstituted intestinal equivalents were constructed using Caco-2, U937, and L929 cells, exactly as described previously by Truzzi et al., (2022) [22]. After a period of 5 days, the models were treated with the combination treatment (1.2 µM mM SPD + 0.2 mM EUG) and SUPPL (1.2 µM SPD and 92 µM EUG), respectively, or DMEM for the untreated control and LPS treatment alone for 1 h. Thereafter, LPS (1 ng/mL) was added to the LPS treatment, as well as the two experimental treatments for a further 24 h. For the immunohistochemical analyses, the cells were fixed with formalin for 2 h at room temperature, dehydrated, and embedded in paraffin. Paraffin-embedded reconstituted 3D intestinal equivalents were then rehydrated, and sections (4 μm thick) were stained with H&E. For LC3-II and occludin detection, the cells were labeled with LC3-II (Novus) antibody and occludin (Novus) antibody, respectively. Immunohistochemistry was performed using fast red chromogen UltraTek Alk-Phos Anti-Polyvalent (permanent red) staining kit according to the manufacturer’s instructions. Negative controls were obtained by omitting the primary antibody. The slides were examined under the microscope (MEIJI Techno Co., Ltd.) at a magnification of ×40. Acidic mucin and mucopolysaccharides produced by the untreated control, the LPS treatment, and the two SPD and EUG experimental treatments, respectively, were determined by Alcian blue and periodic acid-Schiff (PAS) staining kits according to the manufacturer’s instructions. Micrographs were taken on a Confocal Scanning Laser Microscopy at a magnification of ×60. The percentage of positive pixels was analyzed and described in Section 4.7.

### 4.11. Statistical Analysis

Data from chemical analyses were represented as mean values. The cell tests were performed in triplicate and the data expressed as mean values of the three different experiments. Statistical analysis was conducted using GraphPad Prism Version 9.3.1 (2021). The one-way variance (ANOVA) was used to determine any significant differences between treatments. Using Dunnett’s multiple comparisons test, significant differences were represented as follows: ns *p* ≥ 0.05, * 0.01 < *p* < 0.05, ** 0.001 < *p* < 0.01, *** 0.0001 < *p* < 0.001, **** *p* < 0.0001. In the graphs, mean values expressed with stars were statistically different.

## 5. Conclusions

The SUPPL containing a combination of SPD and EUG was not shown to be toxic to the Caco-2 and NCM460 intestinal cell lines. Under inflammatory conditions induced by LPS, the SUPPL ameliorated ROS levels and MDK inflammatory marker expression. Moreover, the SUPPL attenuated occludin expression and mucus production under LPS challenge in the 3D reconstituted intestinal models. Over a short timeline of 2–4 h, the SUPPL was shown to significantly stimulate LC3-II and P62 steady state expression and turnover in both the presence and absence of LPS-induced inflammation compared to the untreated control and LPS-treated cells alone. Using the early autophagy blocker dorsomorphin, inflammatory MDK levels were reduced in an autophagy-independent manner in SUPPL + LPS-treated NCM460 cells. Over a timeline of 24 h, the mitophagy receptor BNIP3L was suppressed in the SUPPL + LPS cells compared to the presence of significant expression in the LPS-treated cells alone. Instead, the SUPPL + LPS-treated cells showed an increased expression of conventional autophagy protein compared to the untreated control and LPS-treated cells alone. Autophagy flux studies and the role of autophagy in modulating inflammation over longer timelines of 24 h will be the subject of future research. The SUPPL was shown to be effective in reducing inflammation as well as stimulating autophagy and can be recommended along with a healthy diet in maintaining intestinal cell health.

## Figures and Tables

**Figure 1 ijms-24-04131-f001:**
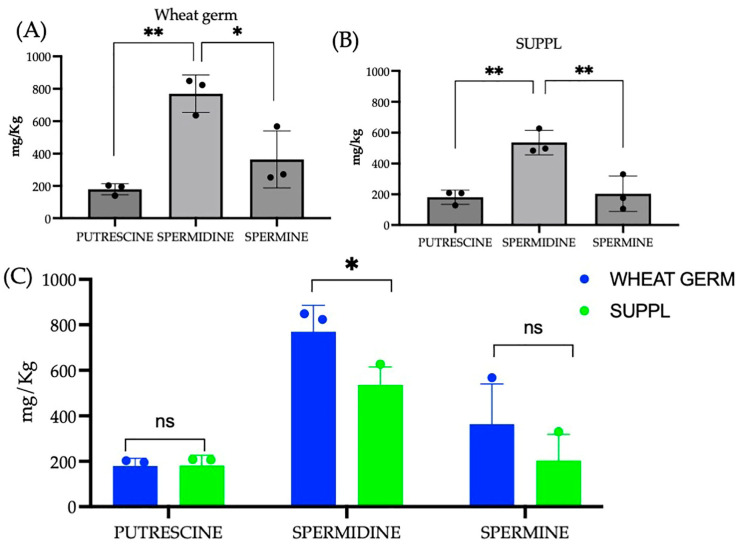
Polyamine content (putrescine, spermidine and spermine) in both the pressed wheat germ source (**A**) and the supplement (**B**) with a statistical comparison between the pressed wheat germ and supplement (**C**). Significant differences were represented as follows: ns *p* ≥ 0.05, * 0.01 < *p* < 0.05, ** 0.001 < *p* < 0.01. The black dots indicate the positioning of the individual replicates within the bar for each sample.

**Figure 2 ijms-24-04131-f002:**
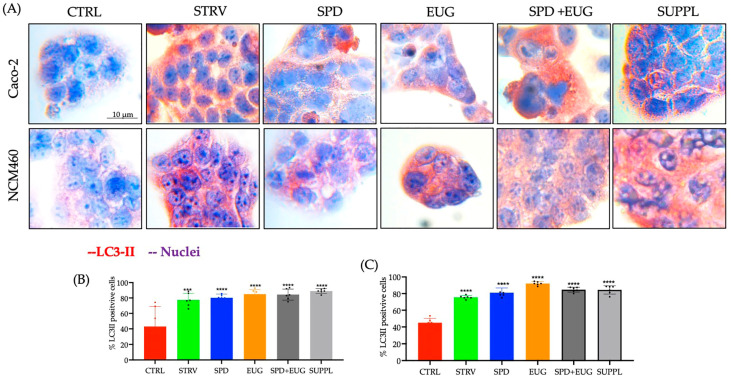
The effect of 1.2 µM spermidine (SPD) and 0.2 mM eugenol (EUG) alone, in combination (1.2 µM SPD + 0.2 mM EUG), and in supplement (SUPPL) form (1.2 µM SPD + 92 µM EUG), respectively, on LC3-II activation in Caco-2 (**A**,**B**) and NCM460 (**A**,**C**) cell lines compared to the untreated control (CTRL). (**A**–**C**) LC3-II expression in the CTRL, the starvation (STRV)-induced control, and all treatments was detected with red chromogen staining after 4 h and visualized at ×40 magnification. (**B**,**C**) Statistical analysis of the LC3-II positive cells stained with red chromogen was performed with significant differences represented as follows: *** 0.0001 < *p* < 0.001, **** *p* < 0.0001. The black dots indicate the positioning of the individual replicates within the bar for each sample.

**Figure 3 ijms-24-04131-f003:**
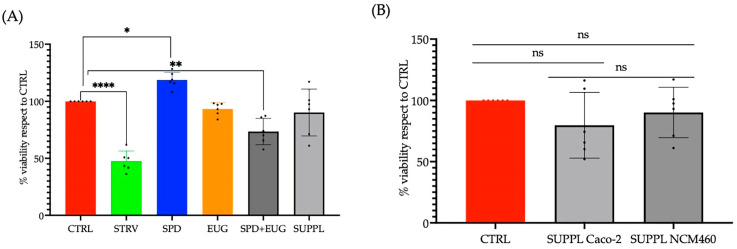
(**A**) The effect of 1.2 µM spermidine (SPD) and 0.2 mM eugenol (EUG) alone, in combination (1.2 µM SPD + 0.2 mM EUG), and in supplement (SUPPL) form (1.2 µM SPD + 92 µM EUG), respectively, on cell viability (MTT assay) in NCM460 cells compared to the untreated control (CTRL). (**B**) The effect of the SUPPL on cell viability in Caco-2 and NCM460 cells compared to the respective controls. Significant differences were represented as follows: ns *p* ≥ 0.05, * 0.01 < *p* < 0.05, ** 0.001 < *p* < 0.01, **** *p* < 0.0001. The black dots indicate the positioning of the individual replicates within the bar for each sample.

**Figure 4 ijms-24-04131-f004:**
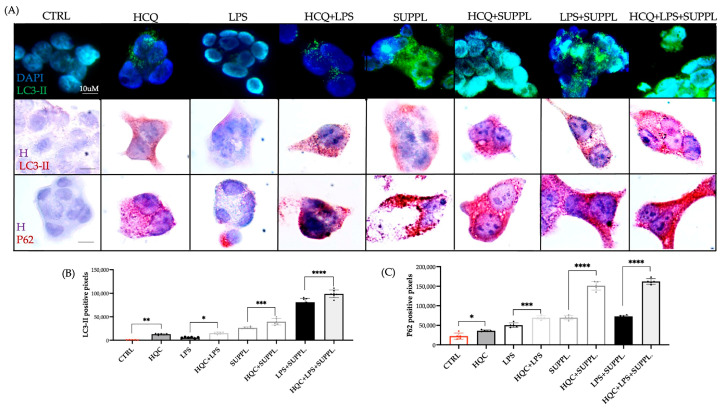
(**A**) The effect of the supplement (SUPPL, 1.2 µM SPD + 92 µM EUG) alone or in combination with ng/mL lipopolysaccharide (LPS) for 2 h on fluorescent green (FG)-stained LC3-II puncta, chromogen red-stained LC3-II and chromogen red-stained P62 in comparison to the untreated control and LPS alone (magnification ×60). All treatments were performed in the presence and absence of 20 µM hydroxychloroquine (HCQ). (**B**) Quantification of the LC3-II steady state and turnover levels. (**C**) Quantification of the P62 steady state and turnover levels. Significant differences were represented as follows: * 0.01 < *p* < 0.05, ** 0.001 < *p* < 0.01, *** 0.0001 < *p* < 0.001, **** *p* < 0.0001. The black dots indicate the positioning of the individual replicates within the bar for each sample. DAPI = nuclear staining in FG imaging: H = hematoxylin.

**Figure 5 ijms-24-04131-f005:**
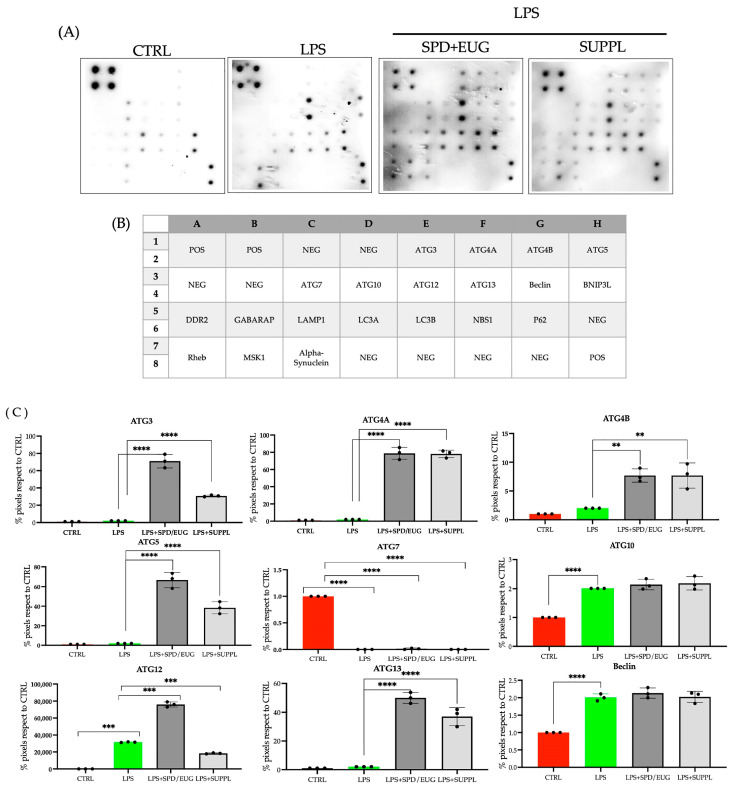
Expression (**A**) of 20 individual human autophagy proteins in NCM460 cells, identified according to the (**B**) position location on the micro-array. Protein expression (**A**) was performed in the untreated control (CTRL) and lipopolysaccharide (LPS)-treated cells (1 ng/mL LPS) after 24 h and compared to cells pre-incubated for 1 h with 1.2 µM spermidine (SPD) + 0.2 mM eugenol (EUG) and in supplement (SUPPL) form (1.2 µM SPD + 92 µM EUG) prior to LPS addition. (**C**) Protein expression for each of the 20 proteins was quantified for each treatment. Significant differences were represented as follows: * 0.01 < *p* < 0.05, ** 0.001 < *p* < 0.01, *** 0.0001 < *p* < 0.001, **** *p* < 0.0001. The black dots indicate the positioning of the individual replicates within the bar for each sample.

**Figure 6 ijms-24-04131-f006:**
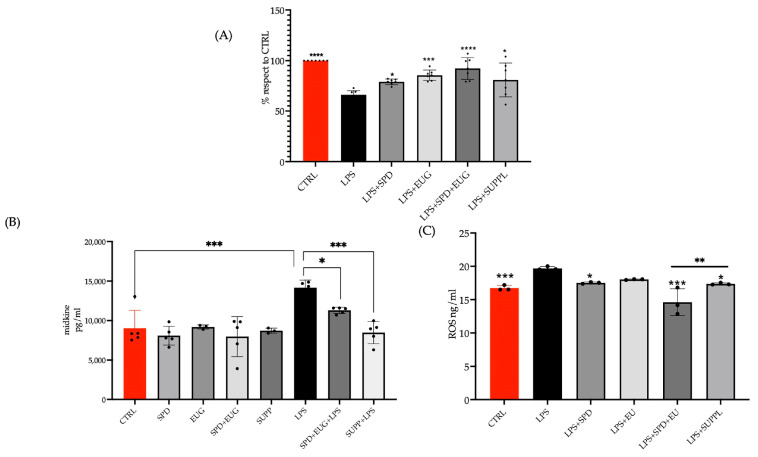
The effect of a pretreatment (1 h) of 1.2 µM spermidine (SPD) and 0.2 mM eugenol (EUG) alone, in combination (1.2 µM SPD + 0.2 mM EUG), and in supplement (SUPPL) form (1.2 µM SPD + 92 µM EUG) prior to the addition of 1 ng/mL lipopolysaccharide (LPS) for 24 h on (**A**) cell viability, (**B**) Midkine levels, and (**C**) ROS levels in NCM460 cells compared to the untreated control (CTRL) and LPS treatment alone. Significant differences were represented as follows: * 0.01 < *p* < 0.05, ** 0.001 < *p* < 0.01, *** 0.0001 < *p* < 0.001, **** *p* < 0.0001. The black dots indicate the positioning of the individual replicates within the bar for each sample.

**Figure 7 ijms-24-04131-f007:**
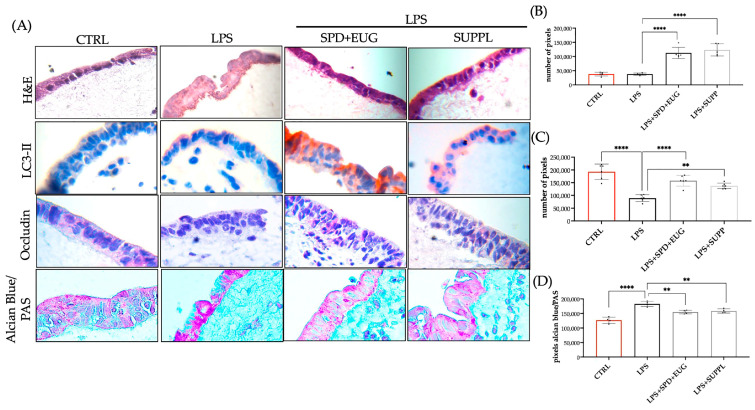
(**A**) Hematoxylin and eosin (H&E) staining, and LC3-II, occludin, and mucus expression of 3D intestinal equivalents (Caco-2/U937/L929 co-cultures) exposed to a 24 h pretreatment of either 1.2 spermidine (SPD) and 0.2 mM eugenol (EUG) or supplement (1.2 µM SPD + 92 µM EUG) prior to the addition of lipopolysaccharide (LPS, 1 ng/mL) for a further 24 h compared to the untreated control (CTRL) and LPS treatments alone. The magnification was ×60. (**B**) Quantification of LC3-II-red chromogen staining, (**C**) occludin–red chromogen staining (**D**) mucus expression from Alcian Blue/Periodic Acid-Schiff Stain (PAS) staining of the Caco-2 cells. Significant differences were represented as follows: ** 0.001 < *p* < 0.01, **** *p* < 0.0001. The black dots indicate the positioning of the individual replicates within the bar for each sample.

**Figure 8 ijms-24-04131-f008:**
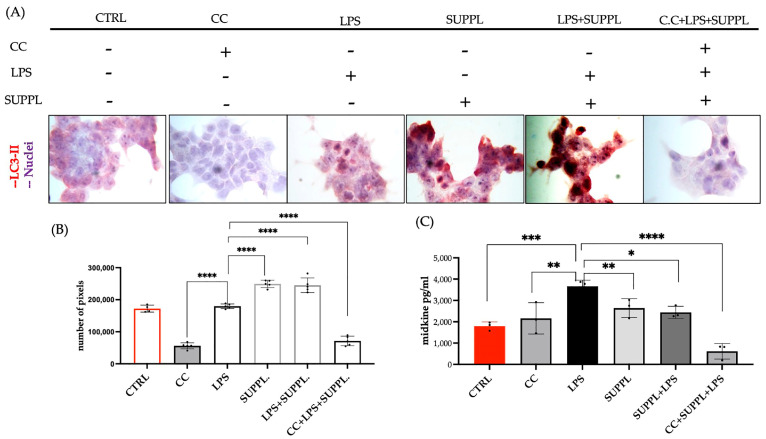
(**A**) Exposure of NCM460 cells to the supplement (SUPPL, 1.2 µM spermidine (SPD)) + 92 µM eugenol (EUG), lipopolysaccharide (LPS, 1 ng/mL) and Compound C (CC, 5 µM) either alone or in combination compared to the untreated control (CTRL). (**A**) Red chromogen-stained LC3-II expression at ×40 mag with (**B**) LC3-II expression quantification and (**C**) comparison to midkine levels. Compound C was administered 30 min prior to the addition of the SUPPL and SUPPL pretreatments were for 1 h prior to LPS exposure for 4 h. Significant differences were represented as follows: * 0.01 < *p* < 0.05, ** 0.001 < *p* < 0.01, *** 0.0001 < *p* < 0.001, **** *p* < 0.0001. The black dots indicate the positioning of the individual replicates within the bar for each sample.

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
