# Peer review of "Spermidine-Eugenol Supplement Preserved Inflammation-Challenged Intestinal Cells by Stimulating Autophagy"

_ijms, 2023, doi:10.3390/ijms24044131_

Round 1

Reviewer 1 Report

Spermidine-eugenol supplement preserved inflammation-challenged intestinal cells by stimulating autophagy by Truzzi et al describes an in vitro study of inducing autophagy to alleviate inflammation in human cell lines with a supplement from wheat germ and clove eugenol. The manuscript describes the production of the supplement briefly and the content of active ingredients in final supplement. Supplement is then tested in autophagic activation, inflammation reduction and ROS reduction under challenge (LPS). Additionally, 3D co-culture experiments were described and the supplement tested for maintenance of cellular integrity and decrease of mucus production. The authors conclude the supplement to be safe in stimulating autophagy in human intestinal cell lines showing promise for human consumption.

The weakness of this study is in the lack of discussion on the necessity of this supplement compared to improved dietary habits and existing products. It should be motivated how much gain the product brings in comparison with consumption of whole grain products as well as existing wheat germ supplements. Experimental wise the manuscript does not describe autophagic flux in a satisfactory manner. While microarray through the process key proteins might be indicative of induced autophagy the standard in the field continues to be comparison of accumulating lipidated LC3 under free flow versus under inhibition of the process at end point (late autophagy inhibitors). Increase in lipidated LC3 (as well as LAMP1 and receptors) can be interpreted as a block in autophagic degradation. Blocking autophagy from the beginning of the process (early blockers that act through signaling and hence the formation of autophagosomes) for 24h would definitely kill the cells. In literature there are however plenty of examples of lower doses of blockers used causing a milder effect or using a later stage inhibitor that might be tolerated by the cells (for example chloroquine). 

Given a major revision of the manuscript including an experiment on lipidated LC3 accumulation under autophagy inhibition with a late inhibitor and justification for the necessity of the supplement this work could be suitable for publication.

Specific comments:

-line 134 contentin>content in

-line 157 powered>powdered

-line 158 space after full stop

-lines 161-165: Lipidated LC3 is not accepted as a marker of increased autophagy on its own. Increased lipidated LC3 can indicate a block in autophagy and therefore autophagic flux needs to be measured

-line 170: calling the combined treatment SPD-EUG can lead someone to intuitively think it means SPD minus EUG (would replace with SUPPL)

-Fig.2 where is the starvation control image?

-Fig.2 Caco2 CTRL image, NCM460 CTRL image and NCM460 SPD image collectively seem different than the other images in this figure. Also, Caco2 CTRL seems quite unfocused. The blue in nuclei should match in all images to represent the increase in red LC3 properly. As such, the figure image is not very convincing. A blow-up image could also be added to highlight the result. Possibly more than 6 fields of 40x images would be more convincing.

-scale bars missing from all microscopy images

-Discrepancy between Fig.2 and materials and methods section. Mat&met states that both nuclei and LC3 were coloured red (which would make the assay very difficult since all nuclei staining should be removed from the overall staining to measure LC3 alone) however in the image nuclei look blue (like Hoechst or DAPI or similar dyes look like, and should probably have been used in combination with red LC3 label)?

-It is not explained in the materials and methods how in practice the measurement of the LC3 label was done with ImageJ (especially important when both nuclei and cytosolic staining are same wavelength?), or if the results were expressed as “% LC3 positive cells” what was the definition/cutoff used to define LC3 positive versus negative?

-Figure 3A: the figure does not explain what is measured (in the MTT assay). The y-axes should preferably state “% what respect to ctrl” e.g. “% viability respect to ctrl” or “OD550…" 

-Fig.3: it should be commented on why SPD+EUG treatment (separate treatments) kill the cells to a significant level

-Fig.4: it should be clear from text what kinds of assays were used for midkine and ROS evaluation (written somewhere that ELISA kits were used)

-Increase in LC3, LAMP1 or receptors (p62, BNIP3L/NIX) can also be interpreted as a block in autophagy/mitophagy(hence the accumulation), this is why autophagic flux should be evaluated by using an autophagy blocker instead

-lines 327-330: no such sample in figure (CC>SUPPL>CC) was it supposed to read CC>SUPPL>LPS?

-in discussion many words together without a space 

-ethanol is not a proper fixative and alcohols in general not recommended to be used with membrane structures because it does not preserve them well (for cell monolayers specifically, not necessarily for immunohistochemical staining of tissues), for LC3 detection it would be better to use chemical fixation prior to alcohol and possible to even not use alcohol at all, also in the method explanation a permeabilization was done with triton which is not needed in combination with alcohol since alcohol is a permebilizing agent

Author Response

Reviewer 1

The weakness of this study is in the lack of discussion on the necessity of this supplement compared to improved dietary habits and existing products. It should be motivated how much gain the product brings in comparison with consumption of whole grain products as well as existing wheat germ supplements.

Wheat germ is rich in spermidine and while it is recommended to improve dietary habits (such as the Mediterranean diet with vegetables containing spermidine), the most effective way of attenuating spermidine levels is by ingesting spermidine in concentrated supplement form (wheat-germ based supplements). However, comparing the present supplement to those of existing supplements was not the objective of the study and we feel it would be perceived as a direct conflict of interest. Moreover, the supplement developed in the present study also contains not only spermidine but also eugenol. The objective was to test a supplement that had two ingredients, one being spermidine that is well reputed to increase autophagy and the other being Eugenol that is a well reported anti-inflammatory. Additional motivation for developing a supplement was based on our previous work showing that both components acted in a similar manner (both individually and in combination) on autophagy and inflammation.

In the Introduction, please see the motivation which reads as follows:

Although SPD is well-reported to be contained in various plant-based foods, wheat germ is the richest source of SPD and is the source of various organic SPD supplements on the market today, also used in experiments on autophagy and inflammaging [20].  While it is beneficial to consume a varied SPD-rich diet (such as the Mediterranean diet), SPD supplements have been recommended in conjunction with a varied diet or in instances where SPD may be deficient (such as in the elderly) or lacking in the natural diet [20,21].  However, the potential advantage of combining two components with complementary attributes has been highlighted [3]. As part of an incentive of the European Institute for Innovation & Technology (EIT) FOOD (www.eitfood.eu) projects using natural products to support either the prevention of COVID-19 or in the treatment of individuals with a higher risk for more severe outcomes of COVID-19, our previous research showed that both wheat germ SPD and clove eugenol (EUG), improved autophagy expression and reduced inflammation in vitro [22]. Aside from the anti-viral properties of both components, SPD was selected for reported benefits of increasing autophagy, whereas there is a particular interest in our group for the antioxidant and anti-inflammatory properties of EUG-rich essential oils [20,23]. Supplements similarly afford the ingestion of EUG in a more concentrated form and interest in the use of EUG as a component of natural medicinal ingredients is currently the object of scientific research [24]. As a result, there was an incentive in developing a premade, economically priced, natural, dual component supplement to examine the efficacy on both autophagy and inflammation.

Then in the Discussion:

The SUPPL was shown to be equally efficient in reducing inflammatory parameters and increasing autophagy protein expression as when the two components were added separately (SPD+EUG), showing that in a SUPPL form, the principal ingredients retained their properties. Although, a healthy, varied diet is always recommended, the SUPPL was shown to afford protection to intestinal cells and may be used in conjunction with a healthy diet, but also in diets where SPD and EUG are largely lacking, as well as for individuals with inflammatory issues. Although the present study is preliminary, to best of our knowledge, there are no previous studies using a preprepared dual component SUPPL specifically addressing the relationship between autophagy and inflammation.

Experimental wise the manuscript does not describe autophagic flux in a satisfactory manner. While microarray through the process key proteins might be indicative of induced autophagy the standard in the field continues to be comparison of accumulating lipidated LC3 under free flow versus under inhibition of the process at end point (late autophagy inhibitors). Increase in lipidated LC3 (as well as LAMP1 and receptors) can be interpreted as a block in autophagic degradation.

Blocking autophagy from the beginning of the process (early blockers that act through signaling and hence the formation of autophagosomes) for 24h would definitely kill the cells. In literature there are however plenty of examples of lower doses of blockers used causing a milder effect or using a later stage inhibitor that might be tolerated by the cells (for example chloroquine). 

This is true that we used steady state levels (LC3-II) and increases in protein content (microarray) which are indirect methods for assessing the potential increase in autophagy. We did however notice that with the early blocker, Compound C, autophagy was blocked after 4 h and that this block was associated with no expression in LC3 and no problems with cell viability (Figure 6). Given the comment above that the use of an early blocker such as compound C would obviously kill the cells after 24 h, we have removed this result (Figure 7) from the article and included it in a supplementary Figure

Given that autophagic flux is measured in a direct manner using late autophagy inhibitors, we performed an experiment in which both LC3-II and P62 were measured for each treatment in both the absence of hydroxychloroquine (HCQ) and in the presence of HCQ.

The reviewer is requested to see the results of this experiment in the Results section and to refer to the Discussion.

For a rapid glance – see the new figure below

 Given a major revision of the manuscript including an experiment on lipidated LC3 accumulation under autophagy inhibition with a late inhibitor and justification for the necessity of the supplement this work could be suitable for publication.

We have attempted to address both issues.

Specific comments:

-line 134 contentin>content in

This formatting error has been changed and now reads content in

-line 157 powered>powdered

This error has been changed and reads powdered.

-line 158 space after full stop

The space has been added.

-lines 161-165: Lipidated LC3 is not accepted as a marker of increased autophagy on its own. Increased lipidated LC3 can indicate a block in autophagy and therefore autophagic flux needs to be measured.

This is true. The sentence now reads:

The steady state lipidified protein 1 light chain 3 (LC3-II, or LC3B), produced from the covalent linkage of cytosolic LC3-I to the phosphatidylethanolamine (PE) lipid on the surface of the phagophore, was firstly employed as an indirect measure of increased autophagy.

-line 170: calling the combined treatment SPD-EUG can lead someone to intuitively think it means SPD minus EUG (would replace with SUPPL).

In the Introduction this was written:

Hence, the first objective of the present investigation was to establish safety of the combined SPD and EUG supplement (from hereinafter referred to SUPPL). Thereafter,

SPD-EUG was replaced with SUPPL throughout the document.

Fig.2 where is the starvation control image?

The starvation control images were added. See Figure 2.

-Fig.2 Caco2 CTRL image, NCM460 CTRL image and NCM460 SPD image collectively seem different than the other images in this figure. Also, Caco2 CTRL seems quite unfocused. The blue in nuclei should match in all images to represent the increase in red LC3 properly. As such, the figure image is not very convincing. A blow-up image could also be added to highlight the result. Possibly more than 6 fields of 40x images would be more convincing.

This has been changed

-scale bars missing from all microscopy images

The scale bars have now been added to the images.

-Discrepancy between Fig.2 and materials and methods section. Mat&met states that both nuclei and LC3 were coloured red (which would make the assay very difficult since all nuclei staining should be removed from the overall staining to measure LC3 alone) however in the image nuclei look blue (like Hoechst or DAPI or similar dyes look like, and should probably have been used in combination with red LC3 label)?

This was a mistake for which we apologize. The method was incorrectly written. LC3 was colored with red chromogen and the nuclei colored a deep purple with haematoxylin. Given that the next comment is also related to the detection of LC3 – we include the changes for both points below the next comment.

It is not explained in the materials and methods how in practice the measurement of the LC3 label was done with ImageJ (especially important when both nuclei and cytosolic staining are same wavelength?)

….. or if the results were expressed as “% LC3 positive cells” what was the definition/cutoff used to define LC3 positive versus negative?

The explanation has now been provided and included under the subheading “4.6. Immunocytochemistry for autophagy LC3-II marker detection” as follows:

Cells were permeabilized with 0.1% TritonX100 and stained for LC3-II (Novus) antibody and then labelled with the UltraTek Alk-Phos Anti-Polyvalent (permanent red) staining kit according to the manufacturer’s instructions. Then slides were stained with haematoxylin, which stains the nuclear material deep purple. The slides were examined under the microscope (MEIJI Techno CO., L.T.D.) at a magnification of x40 to identify LC3-II positive cells. The nuclei were quantified, and more than 400 nuclei were counted for each sample. Results are expressed as the percentage of LC3-II positive cells to the number of nuclei counted. Pictures of cells were analyzed by using ImageJ2 software (Wayne Rasband, version 2.9.0/1.53t).  To perform the analysis of the pixels, digital images were processed to 300 pixels/inch and converted to 8 bits. Then, the binary images were subjected to “color deconvolution” plugin to analyze haematoxylin staining and permanent red. The selected picture was saved as a tiff and subjected to a “clean-up” procedure to eliminate artefacts with Adobe PhotoshopCC (version 20.0.4). Then the interested area was measured with the application “Analyze particle” of ImageJ2. Each experiment was performed in triplicate and data reported as the number of pixels.

-Figure 3A: the figure does not explain what is measured (in the MTT assay). The y-axes should preferably state “% what respect to ctrl” e.g. “% viability respect to ctrl” or “OD550…" 

Refer to Figure 3. This now reads: % Viability with respect to the control.

-Fig.3: it should be commented on why SPD+EUG treatment (separate treatments) kill the cells to a significant level.

We don’t know why viability was lower. We thank the reviewer for the observation and we will try to address this aspect in future a work.

-Fig.4: it should be clear from text what kinds of assays were used for midkine and ROS evaluation (written somewhere that ELISA kits were used).

The types of assays have been explained in the methods section. It now reads:

ROS was measured using the Human Reactive Oxygen Species Modulator 1 (RO-MO1) ELISA Kit according to the instructions provided in the kit. In brief, samples were added to wells of a microplate precoated with ROMO1 antibody. ROS present in the sample and standards was then bound by the immobilized antibody, after which an enzyme-linked polyclonal antibody specific for ROMO1 was added. The complex was visualized from the addition of a color-based substrate. The amount of ROS present was proportional to the intensity of the color measured at 450 nm. MDK in the supernatants was measured using the human MDK ELISA kit. The principle employed an antibody specific for Human MDK coated on a 96-well plate. MDK present in the samples and standards was bound to the wells by the immobilized antibody. Biotinylated anti-Human MDK antibody was then added and subsequently bound to HRP-conjugated streptavidin. Finally, a substrate was added, and the color measured at 450 nm. Color intensity was proportional to the amount of bound MDK.

-Increase in LC3, LAMP1 or receptors (p62, BNIP3L/NIX) can also be interpreted as a block in autophagy/mitophagy (hence the accumulation), this is why autophagic flux should be evaluated by using an autophagy blocker instead.

We verified the turnover in autophagy by measuring LC3B accumulation with LPS + HQC over the short-term (2 h). However, this experiment was not performed over 24 h. Nonetheless, the Discussion has been rewritten in order not to make assumptions that are not based on experimental finding and the Reviewer is invited to reread the Discussion.

-lines 327-330: no such sample in figure (CC>SUPPL>CC) was it supposed to read CC>SUPPL>LPS?.

Yes, this was a mistake. It should have read LPS. The sentence now reads:

Addition of the supplement for 1 h to cells that had been incubated with CC for 30 min, followed by a 4 h exposure to LPS showed a decrease in MDK, suggesting that MDK reduction was not directly regulated by autophagy within this timeframe.

-in discussion many words together without a space 

It appears that these errors occurred when pasting the various sections onto the template. We apologize for this and have corrected the problem.

-ethanol is not a proper fixative and alcohols in general not recommended to be used with membrane structures because it does not preserve them well (for cell monolayers specifically, not necessarily for immunohistochemical staining of tissues), for LC3 detection it would be better to use chemical fixation prior to alcohol and possible to even not use alcohol at all, also in the method explanation a permeabilization was done with triton which is not needed in combination with alcohol since alcohol is a permebilizing agent.

Since we were asked to include a late autophagy blocker in the manuscript to measure LC3 turnover, we followed this advice in fixing cells for treatment with FG and red chromogen staining of LC3 and P62. A chemical fixation was implemented prior to the addition of alcohol following the protocol of Schläfli et al., 2015 (entitled “Reliable LC3 and p62 autophagy marker detection in formalin fixed paraffin embedded human tissue by immunohistochemistry”). Refer to the Methods.

However, in our previous article (Truzzi et al., 2022), FG staining of LC3 was also performed, and the cells fixed in alcohol with no detrimental side-effects to the preservation of the cells. The Reviewer may compare the images in that article fixed firstly with alcohol to the images in this manuscript fixed firstly with formalin.

Reviewer 2 Report

1. Are there controversies in this field? What are the most recent and important achievements in the field? In my opinion, answers to these questions should be emphasized. Perhaps, in some cases, novelty of the recent achievements should be highlighted by indicating the year of publication in the text of the manuscript.

2. The results and discussion section is very weak and no emphasis is given on the discussion of the results like why certain effects are coming in to existence and what could be the possible reason behind them?

3. Conclusion: not properly written.

4. Results and conclusion: The section devoted to the explanation of the results suffers from the same problems revealed so far. Your storyline in the results section (and conclusion) is hard to follow. Moreover, the conclusions reached are really far from what one can infer from the empirical results.

5. The discussion should be rather organized around arguments avoiding simply describing details without providing much meaning. A real discussion should also link the findings of the study to theory and/or literature.

6. Spacing, punctuation marks, grammar, and spelling errors should be reviewed thoroughly. I found so many typos throughout the manuscript.

7. English is modest. Therefore, the authors need to improve their writing style. In addition, the whole manuscript needs to be checked by native English speakers.

Author Response

Reviewer 2

Regarding Points 1-5, the Introduction and more specifically the Discussion and Conclusions have been extensively modified to address the comments below. Moreover, the Results section has been reorganized and an additional experiment included. I have left all changes in Blue in the uploaded version.

Comments and Suggestions for Authors

1.Are there controversies in this field? What are the most recent and important achievements in the field? In my opinion, answers to these questions should be emphasized. Perhaps, in some cases, novelty of the recent achievements should be highlighted by indicating the year of publication in the text of the manuscript.

2.The results and discussion section is very weak and no emphasis is given on the discussion of the results like why certain effects are coming in to existence and what could be the possible reason behind them?

3.Conclusion: not properly written.

4.Results and conclusion: The section devoted to the explanation of the results suffers from the same problems revealed so far. Your storyline in the results section (and conclusion) is hard to follow. Moreover, the conclusions reached are really far from what one can infer from the empirical results.

5.The discussion should be rather organized around arguments avoiding simply describing details without providing much meaning. A real discussion should also link the findings of the study to theory and/or literature.

For points 6 and 7, it appears that many errors (particularly spacing and punctuation marks) occurred when pasting the various sections onto the template. We apologize for this and have corrected the problem. The manuscript has been checked by a native speaker for the errors mentioned above.

6. Spacing, punctuation marks, grammar, and spelling errors should be reviewed thoroughly. I found so many typos throughout the manuscript.

7. English is modest. Therefore, the authors need to improve their writing style. In addition, the whole manuscript needs to be checked by native English speakers.

Round 2

Reviewer 1 Report

Spermidine-eugenol supplement preserved inflammation-challenged intestinal cells by stimulating autophagy by Truzzi et al. revised version addresses the biggest weaknesses of the original manuscript in a satisfactory manner, however there are some further issues in the new version which would warrant revision before publication. 

The authors motivate in their rebuttal the use of the supplement and state that it was not the objective to compare it to existing spermidine supplements which is a fair statement. They have added text about the importance of dietary habits (referring to Mediterranean diet and healthy diet) but not for example anything about consumption of wheat germ specifically. It is not clear exactly what they mean when they are talking about diets that lacking spermidine. Do they mean wheat allergy or something else? To the reader of this paper the question still remains, why the need of a supplement when wheat germ and clove can be directly consumed. The authors motivate the use of the supplement as a combination of the two components to a satisfactory level.

The authors have added experiments requested measuring the autophagic flux in the way that is still considered the standard in the field (with the presence of a late inhibitor) and the results support their previous conclusions in a satisfactory manner. The more prominent difference in p62 stain compared with LC3 with/without chloroquine treatment is actually not even surprising considering that as a receptor, p62 is quite rapidly degraded in free flow whereas LC3 remains in the autophagic membrane even after the contents are largely degraded. This experiment significally improved the manuscript. However, some questions arise from the added data visualization and the description of the used quantitation method. Why are the images that are chosen for representation not uniform? Why did the authors choose to have a white background in the previous immunofluorescence images and then black in the new ones? It would give a better overall image of the paper to have a uniform style to present microscopic data. 

Additionally, the new data that was added was done according to a paper (cited by the authors in the rebuttal) which may describe a reliable method to detect autophagy markers in question, but more recent and higher impact international ones might have been available for reference. The cited paper includes figures of microscopic images that give relatively little detail being of low magnification and resolution. While the data behind the representative images is unarguably more important, the representing of the data still gives credibility to the story. Further, the description in the material and methods warrants revising.

The text and discussion part where mitophagy is speculated repetitively writes “receptor-based mitophagy signaling molecule”. It seems like a very complicated way to simply write “mitophagy receptor” which BNIP3L genuinely is. BNIP3L and SQSTM1/p62 are both autophagy receptors that function in the degradation of mitochondria in parallel pathways. This is not exactly clear from the text in the discussion. It is ambiguously written that p62 expression is associated with increased mitophagy. How about the other mitophagy receptors/pathways? Assessing which mitophagy pathways are involved and activated by the supplement would indeed be very interesting but assumingly out of the scope of this paper. Therefore, the authors should be careful when speculating on the topic if not followed through with investigation on the matter. This section could hence be a bit clearer.

Given a minor revision of the manuscript including careful thought on the specific comments, this work could be suitable for publication.

Specific comments:

-Figure 4: uniform style in the pictures would be preferred, also still some of the images seem very light and others are darker, could better representative images be selected?

-Figure 6: it would improve the reading experience if the method(s) used to create this data would be mentioned in the main text, in the present (and previous) version it is only mentioned in the material and methods section

-Figure 7: the text in the graph is extremely small and difficult to read

-Discussion: speculations on mitophagy could be clearer, also “receptor-based mitophagy signaling molecule” can be replaced with “mitophagy receptor”

-The description of quantitation needs to be clearer. There might be a discrepancy at the moment in the text since it states first “results were expressed as the percentage of LC3-II positive cells to the number of nuclei counted” and then “data reported as the number of pixels”. In the figures it seems to be written pixels. But then again in the description of the quantitation it is written “measured with the application “analyze particles” of ImageJ2.” which implies that number of particles/p62 positive dots/LC3 positive dots was counted but the data is basically amount of fluorescence pixels in the cells in the image (if this is a correct interpretation). Does this mean number of pixels per number of cells? It really should be stated in the figures in that case and the description should also be clearer.

-the repetitive use of the word “natural” describing the supplement might give a loaded meaning to health-promoting supplements versus synthetized medication

-manuscript should be thoroughly proofread to eliminate things like “than that those” (written several times in the text), extra articles, extra words, “interested area”

-lanes 351-354: “…result in the expression of pro-inflammarory…” “…that negatively impact the expression of…”

-line 523: “The increase in autophagy proteins was evident in all stages of the autophagic pathway (or in all stages of autophagy)…”

Author Response

Spermidine-eugenol supplement preserved inflammation-challenged intestinal cells by stimulating autophagy by Truzzi et al. revised version addresses the biggest weaknesses of the original manuscript in a satisfactory manner, however there are some further issues in the new version which would warrant revision before publication.

The authors motivate in their rebuttal the use of the supplement and state that it was not the objective to compare it to existing spermidine supplements which is a fair statement. They have added text about the importance of dietary habits (referring to Mediterranean diet and healthy diet) but not for example anything about consumption of wheat germ specifically. It is not clear exactly what they mean when they are talking about diets that lacking spermidine. Do they mean wheat allergy or something else? To the reader of this paper the question remains, why the need of a supplement when wheat germ and clove can be directly consumed. The authors motivate the use of the supplement as a combination of the two components to a satisfactory level.

People with wheat allergies would presumably not be able to consume (or not be inclined to consume) wheat germ, whether it be present in either food or in supplement form. Wheat germ is one of the best sources of SPD, however it is not easy to find wheat germ sold in local supermarkets even in a country such as Italy where many try to adhere to a Mediterranean diet (as an author of this paper, I searched for it in many local supermarkets and, unfortunately, could not find it). Moreover, most wheat-based foods sold today (bread, pizza, pasta) are made from refined flour lacking the germ. So, in effect, it is not so easy for the “average” shopper to consume the germ.

As far as clove goes, although cloves can be purchased from the local supermarket, they are do form part of the dietary consumption in Europe (predominantly consumed in some Asian and African cuisines). Moreover, clove essential oil contains a much higher dose of eugenol than either whole or ground cloves, so it is more effective to consume the oil in supplement form than the direct consumption of the cloves.

In the Introduction, I have attempted to address these aspects:

Although SPD is well-reported to be contained in various plant-based foods, wheat germ is the richest source of SPD and is the source of various non-synthetic SPD supplements on the market today, also used in experiments on autophagy and inflammaging [20]. Given that wheat-based foods available are predominantly made from refined flour or semolina, wheat germ SPD is not readily accessible. While it is beneficial to consume a varied SPD-rich diet (such as the Mediterranean diet), SPD supplements are recommended in conjunction with a varied diet or in instances where SPD may be deficient (such as in the elderly) or where SPD consumption from local foods is low [20-21].

Then:

….whereas there is a particular interest in our group for the antioxidant and anti-inflammatory properties of EUG-rich essential oils [23]. Since clove essential oil contains a much higher dose of EUG than either whole or ground cloves (which are also not widely consumed in European diets), supplements similarly afford the ingestion of EUG in a more concentrated form.

The authors have added experiments requested measuring the autophagic flux in the way that is still considered the standard in the field (with the presence of a late inhibitor) and the results support their previous conclusions in a satisfactory manner. The more prominent difference in p62 stain compared with LC3 with/without chloroquine treatment is actually not even surprising considering that as a receptor, p62 is quite rapidly degraded in free flow whereas LC3 remains in the autophagic membrane even after the contents are largely degraded. This experiment significally improved the manuscript. However, some questions arise from the added data visualization and the description of the used quantitation method. Why are the images that are chosen for representation not uniform? Why did the authors choose to have a white background in the previous immunofluorescence images and then black in the new ones? It would give a better overall image of the paper to have a uniform style to present microscopic data.

In response to the first question, Figure 4 has been corrected rendering the images more uniform. Then to respond to the next questions collectively, the coloration methods were different. Firstly, we performed immunofluorescence staining, in which there is a black background. Secondly, we performed an alkaline phosphatase staining in which there is a white background. We chose to stain using two different methods in order to have greater certainty of the data and not to focus on a single technique, so as to avoid any interpretative bias.

Additionally, the new data that was added was done according to a paper (cited by the authors in the rebuttal) which may describe a reliable method to detect autophagy markers in question, but more recent and higher impact international ones might have been available for reference. The cited paper includes figures of microscopic images that give relatively little detail being of low magnification and resolution. While the data behind the representative images is unarguably more important, the representing of the data still gives credibility to the story. Further, the description in the material and methods warrants revising.

We have tried to modify this

The text and discussion part where mitophagy is speculated repetitively writes “receptor-based mitophagy signaling molecule”. It seems like a very complicated way to simply write “mitophagy receptor” which BNIP3L genuinely is.

The phrase “receptor-based mitophagy signaling molecule” has been substituted with “mitophagy receptor” or referred to simply as BNIP3L throughout the manuscript.

BNIP3L and SQSTM1/p62 are both autophagy receptors that function in the degradation of mitochondria in parallel pathways. This is not exactly clear from the text in the discussion. It is ambiguously written that p62 expression is associated with increased mitophagy. How about the other mitophagy receptors/pathways? Assessing which mitophagy pathways are involved and activated by the supplement would indeed be very interesting but assumingly out of the scope of this paper. Therefore, the authors should be careful when speculating on the topic if not followed through with investigation on the matter. This section could hence be a bit clearer.

Yes this part was ambiguous – we apologize. It was interesting that under LPS conditions there was an increase in BNIP3L, GABARAP, BECLIN and P62, all of which are implicated in mitochondrial degradation albeit via different pathways. BNIP3L was particularly noticeable to us as it was the only receptor attenuated by the SUPPL. However, given that mitophagy was not investigated, it is out of the scope of the paper. For this reason, we will not hypothesize on the potential roles of these proteins. The second reviewer also criticized the discussion of results not verified in this manuscript.

As such, in the Discussion, we have only included this:

Whether the increases in conventional autophagy proteins reflected an increase in protective autophagy remains to be verified using late autophagy blockers.

Of interest, SPD+EUG and the SUPPL, in the presence of LPS, attenuated only the expression of BNIP3L, which was significantly expressed in the LPS treatment alone. BNIP3L is currently recognized as a key regulator of mitophagy [41-43,60-61], and whether the downregulation of this mitophagy receptor was attributable to the SUPPL-induced activation of antioxidant pathways resulting in the reduction of ROS (and inflammatory MDK) under LPS challenge remains to be determined.

Given a minor revision of the manuscript including careful thought on the specific comments, this work could be suitable for publication.

Specific comments:

-Figure 4: uniform style in the pictures would be preferred, also still some of the images seem very light and others are darker, could better representative images be selected?

We tried to make the color more homogeneous, but unfortunately we don't have better images to insert into this manuscript

-Figure 6: it would improve the reading experience if the method(s) used to create this data would be mentioned in the main text, in the present (and previous) version it is only mentioned in the material and methods section

The following (below) has been included in the Results section to improve the reading experience:

For MDK

The human MDK ELISA kit, containing a precoated MDK antibody, was used to bind MDK in the samples, which was then quantified.

For ROS

The Human Reactive Oxygen Species Modulator 1 (ROMO1) ELISA Kit was used to quantify ROS based on the extent of binding to the ROMO1 antibody.

-Figure 7: the text in the graph is extremely small and difficult to read

We have enlarged the text in the graph for Figure 7.

-Discussion: speculations on mitophagy could be clearer, also can be replaced with “mitophagy receptor”

The phrase “receptor-based mitophagy signaling molecule” has been substituted with “mitophagy receptor” throughout the article.

-The description of quantitation needs to be clearer. There might be a discrepancy at the moment in the text since it states first “results were expressed as the percentage of LC3-II positive cells to the number of nuclei counted” and then “data reported as the number of pixels”.

In Figure 2, the LC3-II positive cells were counted and the values were reported as a percentage with respect to the control.

In Figure 4, the pixels were quantified as described in the materials and methods section using Image J (Color Deconvolution plugin).

So two different quantification methods were used.

In the methods, we have rendered this clearer.

4.6. Immunocytochemistry for Autophagy LC3-II Marker detection

.............The slides were examined under the microscope (MEIJI Techno CO., L.T.D.) at a magnification of x40 to identify LC3-II positive cells. The nuclei were quantified (representative of Figure 2), and more than 400 nuclei were counted for each sample. Results were expressed as the percentage of LC3-II positive cells to the number of nuclei counted.

4.7. Immunofluorescence and immunocytochemistry for Autophagy LC3-II and P62 expression and turnover with hydroxychloroquine

The nuclear material was stained purple with haematoxylin. LC3-II and P62 expression on slides stained for immunocytochemistry were quantified as follows. Pictures of cells were analyzed by using ImageJ2 software (Wayne Rasband, version 2.9.0/1.53t). To per-form the analysis of the pixels, digital images were processed to 300 pixels/inch and converted to 8 bits. Then, the binary images were subjected to “color deconvolution” plugin to analyze permanent red staining. The selected picture was saved as a tiff and subjected to a “clean-up” procedure to eliminate artefacts with Adobe PhotoshopCC (ver-sion 20.0.4). Thereafter, the interested area was measured with the application “Analyze particle” of ImageJ2. Each experiment was performed in triplicate and data reported as the number of pixels.

4.10. Autophagy LC3-II marker, occludin and mucus detection in a reconstituted intestinal cell model with Caco-2, U937, and L929 cells

......... The percentage of positive pixels was analyzed and described in section 4.7.

In the figures it seems to be written pixels. But then again in the description of the quantitation it is written “measured with the application “analyze particles” of ImageJ2.” which implies that number of particles/p62 positive dots/LC3 positive dots was counted but the data is basically amount of fluorescence pixels in the cells in the image (if this is a correct interpretation). Does this mean number of pixels per number of cells? It really should be stated in the figures in that case and the description should also be clearer.

In Figure 4, we presented the immunofluorescence stained images, but those images were not quantified. We described the images referring to LC3-II stained puncta. The only images that were quantified were those stained with the permanent red kit, and these images were quantified with image J, using "analyze particles".

In the results we wrote this to render it evident:

To this end, LC3-II was firstly visualized using fluorescent green (FG) imagining in the control, and in response to the administration of LPS, the SUPPL, and LPS+SUPPL, respectively, in both the presence and absence of 20 µM HCQ for a short timeline of 2 h (Figure 4A). In addition, LC3-II, as well as sequestosome-1 (also known as the ubiqui-tin-binding protein P62) turnover in both the presence and absence of HCQ for all treatments was visualized using red chromogen staining (Figure 4A), and then statistically quantified (Figure 4B,C).

Then again further down in the results....

“Red chromogen imaging was then also performed with the objective of quantifying LC3-II expression”.

See previous piont for description of the method.

-the repetitive use of the word “natural” describing the supplement might give a loaded meaning to health-promoting supplements versus synthetized medication

That is true. In the Introduction there were 6 separate uses of the word “natural”. In two cases, we substituted “natural” with “plant-based”, in another instance “natural” was replaced with “non-synthetic” and in two instances it was removed.

Overall, the word natural now appears 3 times in the entire manuscript.

-manuscript should be thoroughly proofread to eliminate things like “than that those” (written several times in the text), extra articles, extra words, “interested area”

The “…than that those…” was evident three times in the Results section and has been modified to read “than those”. We apologize for the errors. Moreover, the Introduction has been also streamlined based on the suggestion of the second reviewer. As a result, the above comments have also been dealt with. The manuscript has been carefully read again.

-lanes 351-354: “…result in the expression of pro-inflammarory…” “…that negatively impact the expression of…”

In addition to the error, the sentence was too long has been rewritten. It now reads:

Since inflammation in intestinal cells is widely reported to result in the expression of pro-inflammatory cytokines that impact negatively on TJ protein expression and mucus production, it was necessary to use more physiologically and structurally relevant in vitro models to investigate these parameters.

-line 523: “The increase in autophagy proteins was evident in all stages of the autophagic pathway (or in all stages of autophagy)…”

This has been corrected. 

Reviewer 2 Report

1. Introduction is too long. Please concise and focus only on the keywords.

2. Discussion can improve again by focusing the significant results.

Author Response

  1. Introduction is too long. Please concise and focus only on the keywords.

This is true. Although the second reviewer ask that we expand on the dietary consumption of the principal ingredients, the Introduction has now been streamlined and the overall length reduced.

  1. Discussion can improve again by focusing the significant results.

We have improved the Discussion by not speculating or hypothesizing on experiments we did not perform. We removed extraneous material. Rather we highlight the certain results in conjunction with the need to perform specific experiments.
